# Probiotic Strains Isolated from an Olympic Woman’s Weightlifting Gold Medalist Increase Weight Loss and Exercise Performance in a Mouse Model

**DOI:** 10.3390/nu14061270

**Published:** 2022-03-17

**Authors:** Wen-Yang Lin, Yi-Wei Kuo, Jia-Hung Lin, Chi-Huei Lin, Jui-Fen Chen, Shin-Yu Tsai, Mon-Chien Lee, Yi-Ju Hsu, Chi-Chang Huang, Yung-An Tsou, Hsieh-Hsun Ho

**Affiliations:** 1Department of Research and Design, Bioflag Biotech Co., Ltd., Tainan 744, Taiwan; wen-yang.lin@bioflag.com.tw (W.-Y.L.); vic.kuo@bioflag.com.tw (Y.-W.K.); jiahung.lin@bioflag.com.tw (J.-H.L.); ryan.lin@bioflag.com.tw (C.-H.L.); juifen.chen@bioflag.com.tw (J.-F.C.); shin-yu.tsai@bioflag.com.tw (S.-Y.T.); 2Graduate Institute of Sports Science, National Taiwan Sport University, Taoyuan City 333, Taiwan; kurtlee0710@gmail.com (M.-C.L.); ruby780202@ntsu.edu.tw (Y.-J.H.); john5523@ntsu.edu.tw (C.-C.H.); 3Department of Otolaryngology-Head and Neck Surgery, China Medical University Hospital, Taichung City 404, Taiwan

**Keywords:** probiotics, weight loss, lean body mass, exercise performance

## Abstract

Obesity is a worldwide health problem. Calorie-restricted diets constitute a common intervention for treating obesity. However, an improper calorie-restricted diet can lead to malnutrition, fatigue, poor concretion, muscle loss, and reduced exercise performance. Probiotics have been introduced as an alternative treatment for obesity. In the present study, we tested the weight loss and exercise performance enhancement effectiveness of probiotic strains of different origins, including four isolated from an Olympic weightlifting gold medalist (*Bifidobacterium longum* subsp. *longum* OLP-01, *Lactobacillus plantarum* PL-02, *Lactobacillus salivarius* subsp. *salicinius* SA-03, and *Lactococcus lactis* subsp. *lactis* LY-66). A high-fat diet (HFD) was used to induce obesity in 16 groups of mice (*n* = 8/group). The mice were administered probiotic supplements at a dosage of 4.1 × 10^9^ CFU/kg/day for 10 weeks. All probiotic supplementation groups showed a significant reduction in body weight and fat mass compared with the HFD group. TYCA06, CS-773, BLI-02, PL-02, bv-77, and OLP-01 were the most effective in facilitating weight loss and fat reduction, which may be due to fatty-acid absorbing activity. PL-02, LY-66, TYCA06, CS-773, and OLP-01 elevated the animals’ grip strength and exhaustive running duration. PL-02, LY-66, and OLP-01 increased tissue glycogen (liver and muscle) levels and muscle capillary density and reduced blood lactate production levels after exercise. In conclusion, OLP-01, PL-02, LY-66, TYCA06, and CS-773 were highly effective in enhancing weight loss and exercise performance. This study should be repeated on humans in the future to further confirm the findings.

## 1. Introduction

Obesity is a worldwide health problem and is correlated with cardiovascular disease, diabetes mellitus, sleep–breathing disorders, certain forms of cancer, nonalcoholic fatty liver disease, an increased risk of disability, and metabolic syndrome [1]. Additionally, people with obesity experience a substantial decrease in knee extensor strength and handgrip strength after calorie-restricted weight loss diets [2]. In adults, obesity is defined as a body mass index (BMI, kg/m^2^) of ≥30 [3]. The major causes of obesity are overconsumption of high-calorie foods, irregular exercise, and genetics [4]. Scientists recently discovered that changes in microbial diversity and composition can increase the risk of obesity [5]. People with obesity usually present a relatively high Firmicutes–Bacteroidetes ratio in the gut [6].

Moreover, probiotics are generally considered to be a safe dietary supplement [7] that can facilitate weight loss, reverse microbial dysbiosis-induced obesity, and improve gut integrity [8,9]. For example, a clinical study demonstrated that probiotic strains of *Lactobacillus* and *Bifidobacterium* could significantly reduce BMI, body weight, and the waist–hip ratio compared with a placebo, but they exerted no significant effects on blood lipids, blood sugar, or fat [10]. Furthermore, probiotic strains isolated from an Olympic weightlifting gold medalist, namely, *Bifidobacterium longum* subsp. *longum* OLP-01, *Lactobacillus plantarum* PL-02, and *Lactobacillus salivarius* subsp. *salicinius* SA-03, have been reported to enhance exercise performance [11,12,13]. Huang et al. further observed that *B. longum* OLP-01 could reduce body weight, the epididymal fat pad (EFP), perirenal fat, triglyceride (TG) levels, total cholesterol (TC) levels, and low-density lipoprotein (LDL) levels in mice with obesity induced by a high-fat diet (HFD) [14].

Furthermore, diet and exercise were demonstrated to positively change the gut microbiota by altering transient stool time, reducing contact frequency between the gastrointestinal mucus layer and the pathogens, and protecting the integrity of the intestine [15]. The long-term exercise regimen and strict diet of the Olympic weightlifting athlete may explain why their gut microbial species exhibited a more beneficial effect than those of nonprofessional athletes. However, the mechanisms through which probiotic strains isolated from an Olympic medalist facilitate exercise performance and energy metabolism have yet to be fully elucidated, and the performance of those strains has not been compared with those from a normal healthy human gut. To fill this research gap, the present study tested the effectiveness of probiotic strains of different origins, including four isolated from an Olympic weightlifting gold medalist (OLP-01, PL-02, SA-03, and *Lactococcus lactis* subsp. *lactis* LY-66), in enhancing weight loss in an HFD mouse model. The study findings can contribute to identify probiotic strains that can potentially improve human health.

## 2. Materials and Methods

### 2.1. Probiotics

We obtained 14 probiotic strains—namely, *Lactobacillus acidophilus* TYCA06, *Lactobacillus casei* CS-773, *Bifidobacterium longum* subsp. *infantis* BLI-02, PL-02, *Lactobacillus rhamnosus* bv-77, OLP-01, LY-66, *Bifidobacterium bifidum* SLIM-02, *Streptococcus thermophilus* SY-66, *Lactobacillus salivarius* subsp. *salicinius* AP-32, *Lactobacillus rhamnosus* MP108, *Lactobacillus plantarum* LPL28, SA-03, and *Bifidobacterium animalis* subsp. *lactis* CP-9—from Bioflag Biotech Co., Ltd. (Tainan, Taiwan). Of these strains, OLP-01, PL-02, SA-03, and LY-66 were isolated from the gut of a weightlifting Olympic gold medalist. TYCA06 and SLIM-02 were isolated from healthy human gut, whereas CS-773 and MP108 were isolated from healthy infant gut. Moreover, BLI-02, bv-77, and CP-9 were isolated from human breast milk. SY-66 was isolated from fermented milk, and LPL28 was isolated from Japanese miso. *Lactobacillus plantarum* TWK-10 was purchased from Synbio Technologies (Kaohsiung, Taiwan), and *Lactobacillus rhamnosus* LGG was purchased from Chr. Hansen A/S (Hoersholm, Denmark; Table 1). The *Bifidobacterium* strains were cultured on de Man, Rogosa, and Sharpe (MRS) agar plates (110660, Merck, Darmstadt, Germany) supplemented with 0.05% cysteine and anaerobically incubated at 37 °C for 48 h. The *Lactobacillus* strains were cultured on MRS agar plates and incubated under facultative anaerobic conditions at 37 °C for 48 h.

### 2.2. Animals and Experimental Design

We included 144 Institute of Cancer Research (ICR) strain mice aged 6 weeks (BioLASCO, Yilan, Taiwan) in our experiment. All mice were reared in cages under specific pathogen-free conditions and maintained on a 12 h light/dark cycle at 23 ± 2 °C and 50–60% humidity. We allocated four mice to each cage and supplied them with a sufficient chow diet (No. 5001; PMI Nutrition International, Brentwood, MO, USA) and sterilized water ad libitum. A veterinarian monitored animal behavior and health status. All mice were allowed to acclimatize to the experimental conditions for 2 weeks. The average body weight of the mice was approximately 31 g, and all mice had a similar body weight at the beginning of the experiment.

After the 2 week acclimation period, we randomly arranged the 144 mice into 18 groups (eight mice per group): a low-fat diet (LFD) control group, an HFD control group, and 16 HFD groups fed with 16 different probiotic treatments. The LFD group was fed a normal chow diet. The 17 HFD groups were fed a diet comprising 49.8% chow, 35% lard (I-Mei Foods Company, Taipei, Taiwan), 15% fructose (Fonen Fructose Syrup, Taipei, Taiwan), and 0.2% cholesterol (MP Biomedicals, Santa Ana, CA, United States). The high-fat diet induced obesity animal model followed a previous study [16]. The probiotic groups received probiotic supplementation for 10 weeks. The dose of the 16 probiotic treatment groups (4.1 × 10^9^ CFU/kg) was based on the dose used in a previous study [13]. After 4 weeks of intervention, the forelimb grip strength and exhaustive running time of the 18 groups were measured. At week 10, all mice were sacrificed, and their body fat percentage, blood biochemical factors, and glycogen content were examined (Appendix A). All animal experiments in this study were approved by the Institutional Animal Care and Use Committee (IACUC) of National Taiwan Sport University and conformed to the IACUC-10717 protocol. All authors complied with the ARRIVE guidelines.

### 2.3. Evaluation of Exercise Performance

Four weeks after the probiotic intervention, all groups were subjected to several exercise ability tests. During the testing period (weeks 4 to 5; Appendix A), the probiotic supplements were fed to the mice by oral gavage with a flexible tube at 9:00 a.m. every day. The animals’ body weight, feed intake, and water intake were recorded.

#### 2.3.1. Forelimb Grip Strength

To understand whether probiotic supplementation can improve muscle strength, a forelimb grip strength test was conducted on the 29th day (30 min after feeding with probiotics). The grip strength test was performed using a low-force testing system (Model-RX-5; Aikoh Engineering, Nagoya, Japan), in accordance with previously described procedures [17].

#### 2.3.2. Exhaustive Running Test

An endurance performance test was conducted on the 35th day (30 min after feeding with probiotics). The animals were first trained on and allowed to adapt to a motor-driven treadmill (MK680C, Muromachi Kikai Co. Ltd., Tokyo, Japan). The starting speed was 10 m/min, and the slope was 5%. After 5 min, the speed was increased by 2 m/min until the mouse fell into the shock zone multiple times or was unable to continue forward in the shock zone for more than 5 s. The experimental protocol was the same as that in a previous study [18].

#### 2.3.3. Swimming Test

A swimming test was used to analyze the serum metabolite levels of lactate, blood urea nitrogen (BUN), NH_3_, creatine kinase (CK), and glucose. On the 33rd day, 0.1 mL of blood was collected minutes after feeding with probiotics. Subsequently, all animals were placed in a swimming box filled with water with a temperature of 27 ± 1 °C and swam without weights for 10 min. After 20 min of rest, a total of 0.2 mL of blood was collected at three timepoints for analyzing metabolite levels. Lactate, BUN, CK, serum ammonia, and glucose levels in the blood were tested using an automatic blood analyzer (Hitachi 7060, Hitachi, Tokyo, Japan). The analytical protocol was the same as that in a previous study [19].

### 2.4. Glycogen Levels in the Muscle and Liver

At the end of the experiment, the mouse liver and hindlimb soleus muscles were collected, washed with saline, dried, and weighed. The processed tissue was stored at −80 °C for subsequent analysis of tissue glycogen content. We followed a previously reported method of directly quantifying glycogen [20]. A homogenizing solution was added to the processed tissue sample (5× the volume (*w/v*) of the tissue), and tissues were homogenized using a Bullet Blender homogenizer (Next Advance, Cambridge, MA, USA). The tissue homogenate was dispensed into microcentrifuge tubes and was centrifuged at 4 °C and 12,000× *g* for 15 min. The glycogen content of the upper extract was directly analyzed [20]. Commercially available glycogen standards (Sigma, St. Louis, MO, USA) were used to establish the calibration line to calculate the changes in glycogen levels in the liver and muscle tissues.

### 2.5. Body Fat, Blood Lipids, and Biochemical Variables

After 10 weeks of intervention, the mice were given to the Taiwan Mouse Clinic (Academia Sinica, Taipei, Taiwan) for body fat and body composition assessment using a noninvasive magnetic resonance imaging body composition analyzer (Minispec LF50 TD-NMR, Bruker, Billerica, MA, USA) with a measuring frequency of 7.5 Hz. The biochemical variables in serum samples were tested using an automatic blood analyzer (Hitachi 7060, Hitachi, Tokyo, Japan).

### 2.6. Tissue Sectioning and Histology

The relevant visceral organs, including the heart, liver, kidneys, lungs, gastrocnemius muscle, EFP, and brown adipose tissue (BAT), were excised and weighed. Tissue sections (4 µm) were cut from 10% formalin-fixed paraffin-embedded sample blocks. Section blocks were soaked in xylene and then stained in hematoxylin for 3 min and counterstained with eosin for 1 min. The section block processing procedure was the same as that in a previous study [19].

### 2.7. Fatty-Acid Accumulation in Intestinal Caco-2 Cells

Human colon cancer Caco-2 cells (ATCC HTB-37) were cultured with Dulbecco’s modified Eagle’s medium (DMEM; high glucose, HyClone, Logan, UT, USA) supplemented with 10% fetal bovine serum, 1% penicillin–streptomycin solution (HyClone), and 0.01 mg/mL human transferrin in a 9 cm cell culture dish at 37 °C under 5% CO_2_ for 3 to 5 days. The Caco-2 cells (2 × 10^6^) were subcultured into a six-well plate with the same culture medium for 3 days. The upper layer of the transwell chamber (SPL, Pochon, Korea), which contained a 0.4 μm filtering pore, was placed on the six-well plate; 100 μL of the probiotic solution (2 × 10^8^ CFU/mL) was added to MRS medium with DMEM containing 2% bovine serum albumin and 500 µM oleic acid (OA; Sigma-Aldrich, St. Louis, MO, USA). Two nonprobiotic control groups were cultured: one with 500 μM OA and one without. The upper layer of the transwell chamber was removed, and 0.5% oil red O (dissolved in isopropanol) was mixed with water at a ratio of 3:2 for 15 min. The absorbance value (OD520; in nanometers) was measured in each well. The OD value of the OA control group was defined as 100%. This experimental protocol was the same as that in a previous study [21]. All experiments were conducted in triplicate.

### 2.8. Statistical Analysis

All values are expressed as the mean ± SD. The D’Agostino–Pearson normality test was used to justify the normality of the data (Graphpad Prism 8, Graphpad Software, San Diego, CA, USA). Normality testing results showed that the data in each group were normal (probabilities > 0.05). One-way analysis of variance was performed using SAS software (Cary, NC, USA), and Duncan’s test was used to determine whether differences existed between different treatments. Statistical significance was defined as *p* < 0.05.

## 3. Results

### 3.1. Healthy Human Gut Strain CS-773 and Gold Medalist Strains OLP-01, PL-02, and LY-66 Facilitated Excellent Grip Strength Performance in HFD-Fed Mice

At the beginning of the experiment, the average weight of the mice was approximately 31 g, and no significant differences were observed between the groups (Appendix A). At week 4, testing was conducted to determine whether the probiotics improved exercise performance (Appendix A). Grip strength was measured and calibrated by individual body weight (Figure 1a). The 16 probiotic-fed groups exhibited increased grip strength relative to the HFD control group (*p* < 0.001). The CS-773, OLP-01, and LY-66 groups exhibited excellent grip strength compared with the LFD group (*p* < 0.001). The PL-02, SLIM-02, TYCA06, SA-03, bv-77, BLI-02, and TWK10 groups also demonstrated significantly improved grip strength compared with the control groups (*p* < 0.05; Figure 1a). CS-773 originated from a healthy human gut, and OLP-01, LY-66, and PL-02 were isolated from the gold medalist (Table 1). Moreover, the CS-773 and PL-02 groups exhibited high-density hindlimb muscle fibers in histological sections (Figure 2a).

All probiotic-fed groups, except for the TWK10 group, showed significant improvements in the running test relative to the control groups. The LPL28, PL-02, LY-66, TYCA06, CS-773, BLI-02, and MP108 groups exhibited longer running endurance than the LFD or HFD control groups (*p* < 0.001; Figure 1b); of these strains, PL-02 and LY-66 were isolated from the gold medalist.

### 3.2. Healthy Human Gut Strains TYCA06 and CS-773 and Gold Medalist Strain PL-02 Alleviated Body Weight Gain in HFD-Fed Mice

After 10 weeks, the body weight and fat mass of the HFD group increased significantly compared with those of the LFD group. However, all probiotic-fed groups exhibited significant weight loss and fat mass loss compared with the HFD control group (Figure 1c,d; Table 2). TYCA06, CS-773, BLI-02, PL-02, bv-77, and OLP-01 were associated with the most weight loss, in that order (Figure 1c). TYCA06, PL-02, bv-77, LGG, BLI-02, and CS-773 were associated with the most fat mass loss, in that order (Figure 1d). Among the top 5 weight loss strains, TYCA06 and CS-773 originated from the healthy human gut and PL-02 originated from the gold medalist (Table 1).

We selected six strains that were associated with excellent grip strength performance and weight loss for subsequent tests: three Olympic medalist-derived strains (OLP-01, LY-66, and PL-02) that were associated with enhanced grip strength, two healthy human gut-derived strains (TYCA06 and CS-773) that were associated with weight loss, and LGG, a well-known probiotic strain that has been demonstrated to benefit intestinal health and body weight control [22].

Pathological changes also confirmed the fat mass loss in the probiotic groups. After 10 weeks of probiotic supplementation, the EFP and BAT exhibited smaller adipocyte cell size and number in the probiotic-fed groups, namely, CS-773, TYCA06, OLP-01, PL-02, LY-66, and LGG, as compared with the HFD control group (Figure 2b,c; Table 2). Besides, the six probiotic strains all significantly reduced the glutamate–oxaloacetate transaminase (GOT), glutamate–pyruvate transferase (GPT), TC, TG, and LDL levels compared with the HFD (Table 3).

### 3.3. CS-773, TYCA06, OLP-01, PL-02, and LY-66 Reduced Fatty-Acid Accumulation in Intestinal Caco-2 Cells

CS-773, TYCA06, OLP-01, PL-02, and LY-66 significantly reduced the OA levels to 91% (*p* < 0.05), 93% (*p* < 0.001), 84% (*p* < 0.001), 87% (*p* < 0.001), and 87% (*p* < 0.001), respectively (Figure 3), compared with the OA control (100%) in the in vitro oleic acid absorption test.

### 3.4. Effects of Probiotics on Glycogen Levels

Liver and muscle glycogen levels were measured for selected strains: PL-02, LY-66, OLP-01, TYCA06, and CS-773. The selected probiotic strains engendered a significant increase in liver glycogen levels compared with the LFD and HFD controls (*p* < 0.001). However, the three gold medalist-derived strains (OLP-01, PL-02, and LY-66) were associated with higher liver glycogen levels than the other probiotics (Figure 4a). In addition, the three gold medalist-derived strains were associated with higher muscle glycogen levels. The OLP-01 group had significantly increased muscle glycogen levels compared with the LFD group (*p* < 0.05). The PL-02 and LY-66 groups exhibited a significant elevation in muscle glycogen content compared with the LFD group (*p* < 0.001) and HFD group (*p* < 0.05; Figure 4b).

### 3.5. Gold Medalist-Derived Probiotic Strains OLP-01, PL-02, and LY-66 Reduced Serum Metabolite Levels of Lactate, BUN, Ammonia, and CK in the Swimming Test

The gold medalist-derived strains PL-02, LY-66, and OLP-01 that facilitated glycogen storage were selected to analyze their effects on exercise metabolite levels after an exhaustive swimming test. During high-intensity anaerobic exercise, organisms rapidly generate energy and energy metabolites including lactate. The lactate metabolite levels were similar among the probiotic-fed groups before swimming but were elevated 10 min after swimming (Table 4). The metabolite levels were also compared between the probiotic-fed groups (LGG, PL-02, LY-66, and OLP-01) and the LFD and HFD control groups; the results revealed that the probiotic-fed groups exhibited lower lactate metabolite levels than did the HFD control group. Similarly, after 20 min of rest, the lactate metabolite levels were similar among the probiotic-fed groups.

Furthermore, the probiotic-fed groups (LGG, PL-02, LY-66, and OLP-01) showed significantly lower lactate production rates than did the HFD control group, but the groups did not differ significantly in terms of clearance rate (Table 4). Serum glucose levels were similar among the probiotic-fed and control groups (Table 4). The probiotic-fed groups exhibited significantly reduced serum BUN, ammonia, and CK levels after the swimming test compared with the HFD control group (Table 4).

## 4. Discussion

Grip strength is commonly measured to evaluate muscle strength and function [23]. This study tested the effectiveness of 16 probiotic strains in enhancing exercise performance. An HFD induced obesity and regressed exercise ability in mice. Four probiotic strains, namely, OLP-01, PL-02, SA-03, and LY-66, were isolated from the gut of a weightlifting Olympic gold medalist (Table 1). However, most of the probiotic strains significantly recovered and enhanced grip strength and exhaustive running duration as compared with the HFD (Figure 1a,b). CS-773, OLP-01, LY-66, PL-02, and SLIM-02 were the top 5 strains in terms of grip strength enhancement (Figure 1a).

Exhaustive running is commonly used as an aerobic exercise training test [24]. In this study, the probiotic-fed groups exhibited improved running duration, with the LPL28, PL-02, LY-66, TYCA06, and CS-773 groups being in the top 5 in terms of running duration (Figure 1b). PL-02, LY-66, TYCA06, CS-773, and OLP-01 were associated with excellent grip strength and running duration, yet the gold medalist strain SA-03 did not rank in the top 5 in grip strength or running duration. However, whether the probiotic strains isolated from other sport-specific elite athletes could also improve exercise performance should be tested and analyzed in the future.

Glycogen stores, especially muscle glycogen stores, constitute a crucial contributor to exercise performance [25]. Supplementation with the probiotic strains LGG, OLP-01, PL-02, LY-66, TYCA06, and CS-773 significantly increased liver glycogen stores compared with the LFD and HFD (Figure 4a). Furthermore, PL-02 and LY-66 (both isolated from the gold medalist) were associated with significantly increased levels of muscle glycogen stores compared with the LFD and HFD (Figure 4b). However, the molecular signaling pathway of glycogen synthesis in probiotic strains is unclear. Gut bacteria secreting butyrate, a short-chain fatty acid (SCFA), were reported to function in regulating glycogen metabolism through the GPR43–AKT–GSK3 molecular pathway [26]. Furthermore, probiotics (*L. acidophilus*) were reported to regulate glycogen content in tissues through synthesis-related genes (GSK-3β and Akt) [27]. Insulin secretion limits the elevation of blood glucose and stimulates glycogen synthesis in skeletal muscle [28]. Exercise with OLP-01 supplementation has also been reported to improve insulin sensitivity, which may be correlated with glycogen synthesis [29]. The molecular signaling pathway of glycogen synthesis in the probiotic strains included in this study should be investigated further.

Blood lactate response is a commonly used variable for evaluating exercise performance [30]. The probiotic strains LGG, OLP-01, PL-02, and LY-66 significantly reduced blood lactate levels and the lactate production rate after the swimming test (Table 4). In a previous study, *B. longum* OLP-01 exhibited a similar function in reducing blood lactate after a swimming test [19]. The probiotic strains that increased muscle density in the mice (Figure 2) may contribute to muscle glycogen storage [31] and a decline in blood lactate after exercise [32]. However, the mechanism through which VO_2max_ interacts with the lactate threshold should be assessed in future research [33]. CK, BUN, and ammonia levels are major indicators of fatigue condition after exercise [34,35]. Serum lactate, ammonia, BUN, and glucose are physical fatigue-associated biomarkers, and CK is a tissue damage marker [36]. In the present study, the three gold medalist-derived strains OLP-01, PL-02, and LY-66 significantly reduced serum BUN, ammonia, and CK levels after the exhaustive swimming test; therefore, they may help in alleviating fatigue after high-intensity exercise (Table 4). However, how probiotic strains regulated protein metabolism in metabolic organs including muscles, liver, kidney and intestine should be investigated in the future. 

Weight management is a crucial intervention for obesity [37]. Reducing unnecessary fat without weakening exercise performance is vital for both athletes and nonathletes. Clifton et al. demonstrated that energy-restricted diets can reduce body weight and body fat and increase glycemic control, but without a reduction in total lean mass [38]. Saris et al. indicated that an LFD could result in a significant decrease in body weight and body fat but not in blood TG, TC, or LDL [39].

In the present animal study, all probiotic strains were able to reduce HFD-induced body weight and body fat (Figure 1c,d and Appendix A) while also significantly reducing high levels of blood TG, TC, and LDL (Table 3). Among the top six strains, TYCA06, CS-773, and PL-02 exhibited excellent effectiveness in reducing body weight and body fat (Figure 1c,d, Figure 2b,c and Table 2). Furthermore, TYCA06, CS-773, and PL-02 shrank the EFP size and increased BAT density, which was visible in tissue sections (Figure 2b,c). Increased BAT density is associated with energy expenditure and weight loss [40]. Besides, safety of testing probiotic strains revealed in sectioning slides of lung, heart, liver and kidney (Appendix A). However, how probiotic strains regulated protein metabolism in metabolic organs including muscles, liver, kidney and intestine should be investigated in the future. 

Lee et al. reported that probiotic *Lactobacillus* can absorb host intestinal fatty acid to decrease weight gain, body fat mass, and hepatic lipid accumulation [21]. CS-773, TYCA06, OLP-01, PL-02, and LY-66 showed a significant OA absorption ability (Figure 3), which may be one possible molecular pathway through which body weight and body fat were reduced in the present study (Figure 5).

*L. acidophilus* TYCA06 was reported to have a glucose consumption ability, which may contribute to body weight and body fat control [41]. Additionally, probiotics have been demonstrated to reduce obesity by secreting SCFAs in the gut. Gut microbiota-secreted SCFAs can activate the G-protein-coupled receptors GPR43, peptide-1, and peptide YY, resulting in glucose homeostasis regulation and energy utilization [42]. Moreover, supplementation with *B. longum* was reported to modulate leptin and increase adiponectin [43]. The molecular mechanism through which the probiotic strains limited weight gain, fat, and blood lipids should be investigated in the future.

## 5. Conclusions

In conclusion, this animal study determined that several probiotic strains could facilitate weight loss and exercise performance, including OLP-01, PL-02, LY-66, TYCA06, and CS-773. Of these, OLP-01, PL-02, and LY-66 were isolated from a gold medalist (Figure 5). The findings of this study could be applied in weight management, lipid metabolism, and exercise for the promotion of human health. Probiotic strains may involve lipid metabolism via absorbing excess fatty acids in the gastrointestinal tract. Clinical experiments including lifestyle questionnaire, diet record, body weight, BMI, health condition, aerobic exercise, anaerobic exercise, serum biochemistry profile, gut microbiota analysis, and metabolomics analysis by taking probiotic strains of OLP-01, PL-02, and LY-66 should be tested in the future.

## Figures and Tables

**Figure 1 nutrients-14-01270-f001:**
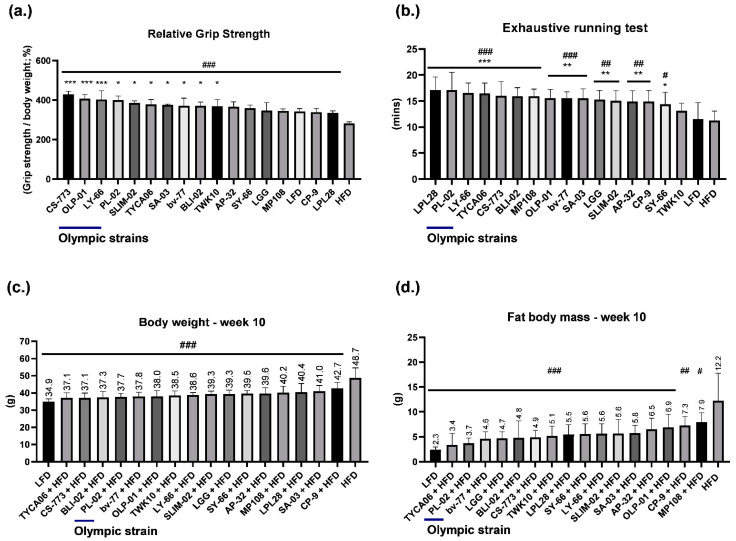
Effects of 10 weeks of probiotic supplementation on (**a**) relative grip strength, (**b**) exhaustive running duration, (**c**) body weight, and (**d**) fat body mass compared with a high-fat diet control group. ^#^
*p* < 0.05, ^##^
*p* < 0.01, ^###^
*p* < 0.001 compared with a high-fat diet control group. * *p* < 0.05, ** *p* < 0.01, *** *p* < 0.001 compared with the low-fat diet group. Data are expressed as mean ± SD for *n* = 8 mice per group. LFD, low-fat diet; HFD, high-fat diet.

**Figure 2 nutrients-14-01270-f002:**
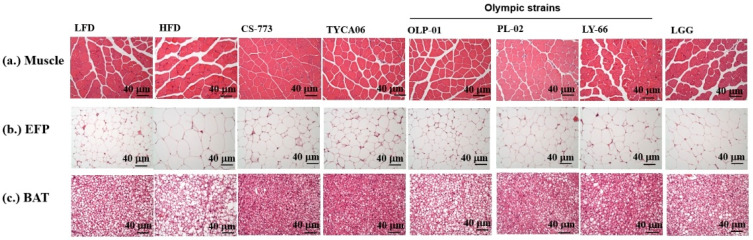
Effects of 10 weeks of probiotic supplementation on histopathology in (**a**) soleus muscle tissue, (**b**) epididymal fat pad (EFP), and (**c**) brown adipose tissue (BAT) in mice. Tissue sections were processed with hematoxylin and eosin stain (magnification, 200×; bar, 40 µm).

**Figure 3 nutrients-14-01270-f003:**
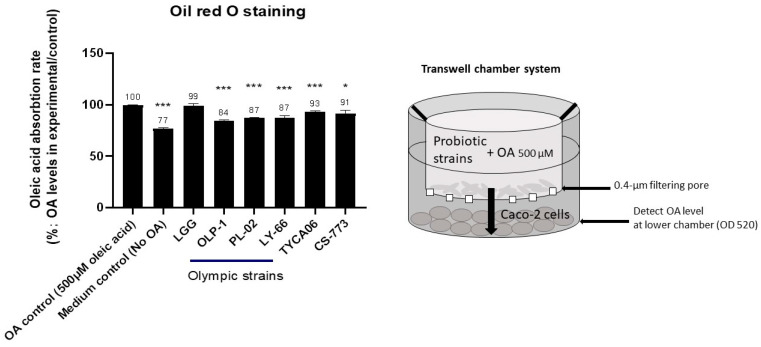
Probiotic strains absorbed fatty acids in vitro. In the experimental group, Caco-2 cells (2 × 10^6^) were seeded to the lower chamber of a six-well transwell, while oleic acid (OA; 500 μM) with probiotics (2 × 10^8^ CFU/mL) was added to the upper chamber of the transwell (OA control: oleic acid without probiotic treatment; medium control: medium without oleic acid or probiotics). Probiotics, OA, and Caco-2 cells were cultured for 3 days, and OA levels in the lower chamber were detected with oil red O stain. OA absorption rate = OA levels in each probiotic group/OA control. * *p* < 0.05 and *** *p* < 0.001 compared with the OA control. All experiments were conducted in triplicate.

**Figure 4 nutrients-14-01270-f004:**
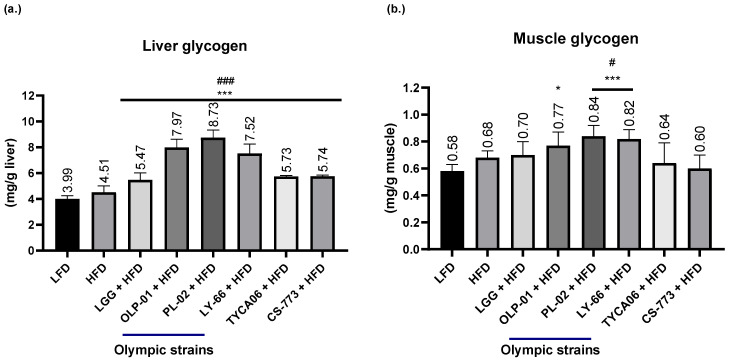
Effects of probiotic treatment on (**a**) liver glycogen levels and (**b**) muscle glycogen levels. OLP-01, PL-02, and LY-66 were isolated from the same Olympic gold medalist. ^#^
*p* < 0.05 and ^###^
*p* < 0.001 compared with the high-fat diet control group. * *p* < 0.05 and *** *p* < 0.001 compared with the low-fat diet group. Data are expressed as the mean ± SD for *n* = 8 mice per group.

**Figure 5 nutrients-14-01270-f005:**
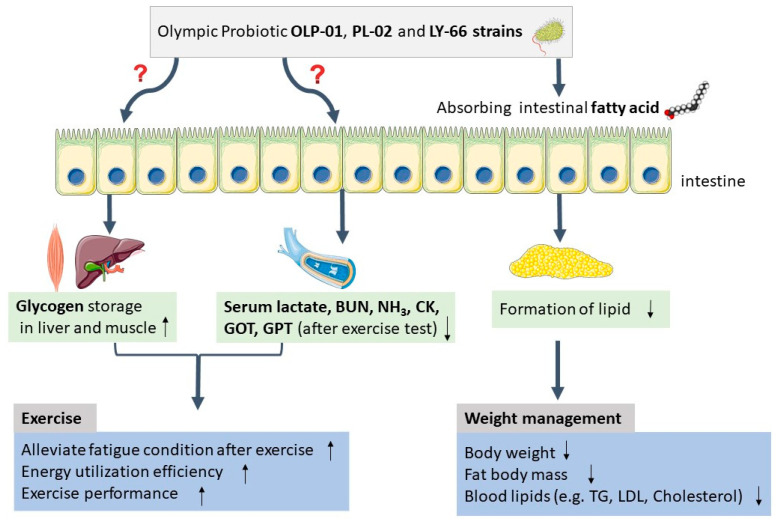
Hypothesis of mechanism through which supplementation with Olympic gold medalist-derived probiotic strains facilitate exercise performance and body weight loss. The upward arrow indicates increasing tendency, whereas the downward arrow indicates decreasing tendency.

**Table 1 nutrients-14-01270-t001:** Origins of probiotic strains.

Strain ID	Species	Origin	Note
OLP-01	*Bifidobacterium longum* subsp. *longum*	Human gut, high-intensity weightlifting athlete	Four strains were isolated from same Olympic athlete
PL-02	*Lactobacillus plantarum*		
SA-03	*Lactobacillus salivarius* subsp. *salicinius*		
LY-66	*Lactococcus lactis* subsp. *lactis*		
TYCA06	*Lactobacillus acidophilus*	Human gut, healthy people	Bioflag Biotech Co., Ltd. (Tainan, Taiwan)
AP-32	*Lactobacillus salivarius* subsp. *salicinius*		
SLIM-02	*Bifidobacterium bifidum*		
LGG	*Lactobacillus rhamnosus*		Chr. Hansen (Hoersholm, Denmark)
CS-773	*Lactobacillus casei*	Human gut, healthy infant	Bioflag Biotech Co., Ltd. (Tainan, Taiwan)
MP108	*Lactobacillus rhamnosus*		
BLI-02	*Bifidobacterium longum* subsp. *infantis*	Healthy human breast	Bioflag Biotech Co., Ltd. (Tainan, Taiwan)
bv-77	*Lactobacillus rhamnosus*		
CP-9	*Bifidobacterium animalis* subsp. *lactis*		
SY-66	*Streptococcus thermophilus*	Fermented food product	Bioflag Biotech Co., Ltd. (Tainan, Taiwan)
LPL28	*Lactobacillus plantarum*	Fermented food product, miso	
TWK-10	*Lactobacillus plantarum*	Fermented food product, Taiwanese Kimchi	SYNBIO TECH INC. (Kaohsiung, Taiwan)

**Table 2 nutrients-14-01270-t002:** Effect of probiotic supplementation on adipose tissue weight.

Treatment	EFP (g)	Perinephric Fat (g)	Mesenteric Fat (g)	BAT (g)
LFD	0.34 ± 0.09 ^###^	0.08 ± 0.01 ^###^	0.68 ± 0.06 ^###^	0.06 ± 0.01 ^###^
HFD	3.06 ± 0.19	1.02 ± 0.13	1.3 ± 0.16	0.14 ± 0.03
TWK10 + HFD	1.23 ± 0.24 ^###^	0.48 ± 0.27 ^###^	0.98 ± 0.25 ^##^	0.08 ± 0.02 ^###^
^$^ OLP-01 + HFD (Olympic strain)	1.17 ± 0.40 ^###^	0.49 ± 0.27 ^###^	0.89 ± 0.14 ^###^	0.09 ± 0.03 ^##^
SLIM-02 + HFD	1.53 ± 0.57 ^###^	0.58 ± 0.29 ^##^	0.96 ± 0.08 ^###^	0.08 ± 0.01 ^###^
LGG + HFD	1.48 ± 0.38 ^###^	0.57 ± 0.22 ^##^	0.78 ± 0.14 ^###^	0.09 ± 0.02 ^##^
^$^ PL-02 + HFD (Olympic strain)	0.82 ± 0.26 ^###^	0.33 ± 0.15 ^###^	0.58 ± 0.18 ^###^	0.09 ± 0.01^##^
^$^ SA-03 + HFD (Olympic strain)	1.835 ± 0.58 ^###^	0.86 ± 0.19	0.98 ± 0.21 ^##^	0.11 ± 0.02 ^#^
CP-9 + HFD	1.69 ± 0.58 ^###^	0.84 ± 0.3	0.93 ± 0.15 ^###^	0.11 ± 0.01
bv-77 + HFD	0.92 ± 0.34 ^###^	0.47 ± 0.24 ^###^	0.76 ± 0.13 ^###^	0.09 ± 0.01 ^###^
AP-32 + HFD	1.46 ± 0.5 ^###^	0.66 ± 0.26 ^#^	0.96 ± 0.28 ^###^	0.1 ± 0.03 ^#^
BLI-02 + HFD	0.98 ± 0.48 ^###^	0.39 ± 0.24 ^###^	0.785 ± 0.23 ^###^	0.09 ± 0.03 ^##^
LPL28 + HFD	1.31 ± 0.2 ^###^	0.68 ± 0.33 ^#^	0.87 ± 0.23 ^###^	0.1 ± 0.02 ^#^
MP108 + HFD	1.89 ± 0.33 ^###^	0.81 ± 0.24	0.97 ± 0.19 ^##^	0.1 ± 0.01 ^#^
TYCA06 + HFD	0.7 ± 0.47 ^###^	0.43 ± 0.26 ^###^	0.77 ± 0.11 ^###^	0.09 ± 0.02 ^##^
^$^ LY-66 + HFD (Olympic strain)	1.15 ± 0.17 ^###^	0.46 ± 0.12 ^###^	0.85 ± 0.06 ^###^	0.1 ± 0.02 ^#^
SY-66 + HFD	1.4 ± 0.35 ^###^	0.65 ± 0.12 ^#^	0.92 ± 0.09 ^###^	0.1 ± 0.02 ^#^
CS-773 + HFD	0.92 ± 0.39 ^###^	0.5 ± 0.26 ^###^	0.76 ± 0.19 ^###^	0.1 ± 0.02 ^#^

^#^*p* < 0.05; ^##^
*p* < 0.01; ^###^
*p* < 0.001 compared with the high-fat diet control group. ^$^ OLP-01, PL-02, SA-03, and LY-66 were isolated from the same Olympic athlete. EFP, epididymal fat pad, BAT, brown adipose tissue.

**Table 3 nutrients-14-01270-t003:** Biochemical evaluation.

Treatment	GOT (U/L)	GPT (U/L)	ALB (mg/dL)	TC (mg/dL)	TG (mg/dL)	HDL (mg/dL)	LDL (mg/dL)	BUN (mg/dL)	Crea (mg/dL)	UA (mg/dL)	TP (mg/dL)	CPK (U/L)	Glu (mg/dL)
LFD	71.12 ± 3.87 ^###^	38.5 ± 5.07 ^##^	3.15 ± 0.12	150.87 ± 11.43 ^###^	57.25 ± 8.97 ^###^	89.26 ± 7.07 ^###^	17.47 ± 2.86 ^###^	14.62 ± 2.52	0.36 ± 0.018	1.32 ± 0.27	5.31 ± 0.23	371 ± 38.3 ^##^	218.12 ± 41.6
HFD	118 ± 6.16	50.87 ± 5.64	3.13 ± 0.07	261.5 ± 18.05	93 ± 11.25	103.56 ± 4.61	74.63 ± 7.3	14.6 ± 3.04	0.36 ± 0.01	1.47 ± 0.5	5.4 ± 0.18	491.75 ± 90.06	259.37 ± 43.16
^$^ OLP-01+ HFD	102.62 ± 6.61 ^###^	40.75 ± 7.24 ^#^	3.12 ± 0.16	201.12 ± 12.59 ^###^	63.87 ± 11.17 ^###^	96.83 ± 8.6 ^#^	49.85 ± 6.56 ^###^	14.13 ± 1.68	0.37 ± 0.02	1.37 ± 0.3	5.36 ± 0.37	490.62 ± 64.67	244.87 ± 29.3
^$^ PL-02 + HFD	99.62 ± 7.63 ^###^	38.75 ± 9.79 ^##^	3.06 ± 0.17	183.12 ± 6.95 ^###^	65.87 ± 15.27 ^###^	95.75 ± 4.5 ^#^	48.81 ± 3.99 ^###^	14.27 ± 2.64	0.36 ± 0.02	1.31 ± 0.2	5.37 ± 0.19	475.87 ± 70.22	248.87 ± 60.17
^$^ LY-66 + HFD	97.75 ± 8.34 ^###^	39.5 ± 9.79 ^##^	3.22 ± 0.18	203.5 ± 15.44 ^###^	66.25 ± 12.15 ^###^	105.57 ± 6.07	48.85 ± 5.32 ^###^	14.03 ± 1.13	0.36 ± 0.02	1.33 ± 0.11	5.53 ± 0.17	462.87 ± 92.29	243.87 ± 29.42
CS-773 + HFD	102.12 ± 5.35 ^###^	37 ± 7.67 ^###^	3.14 ± 0.06	203.37 ± 15.28 ^###^	68.75 ± 12.54 ^###^	96.86 ± 5.28 ^#^	46.77 ± 3.87 ^###^	13.57 ± 1.98	0.36 ± 0.03	1.4 ± 0.24	5.5 ± 0.2	455.25 ± 83.35	241.87 ± 60.76
TYCA06 + HFD	101 ± 7.25 ^###^	41.25 ± 5.89 ^#^	3.1 ± 0.18	190.5 ± 12.43 ^###^	70.12 ± 13.68 ^###^	98.73 ± 9.38	45.57 ± 3.07 ^###^	14.28 ± 2.1	0.36 ± 0.01	1.47 ± 0.42	5.33 ± 0.20	461.5 ± 56.33	241.5 ± 52.24
LGG + HFD	105.5 ± 4.07 ^##^	35.62 ± 5.68 ^###^	3.13 ± 0.06	202.87 ± 13.19 ^###^	62.87 ± 9.59 ^###^	98.23 ± 4.74	54.91 ± 6.56 ^###^	14.56 ± 2.57	0.36 ± 0.01	1.43 ± 0.25	5.31 ± 0.21	483.25 ± 55.59	256.87 ± 30.16

^#^*p* < 0.05; ^##^
*p* < 0.01; ^###^
*p* < 0.001 compared with the high-fat diet control group. ^$^ OLP-01, PL-02, SA-03 and LY-66 were isolated from the same Olympic athlete. GOT, glutamate–oxaloacetate transaminase; GPT, glutamate–pyruvate transferase; ALB, albumin; TC, total cholesterol; TG, triglyceride; HDL, high-density lipoprotein; LDL, low-density lipoprotein; BUN, blood urea nitrogen; Crea, Creatinine; UA, Uric Acid; TP, total protein; CPK, Creatine Phosphokinase; Glu, Glucose.

**Table 4 nutrients-14-01270-t004:** Metabolite profiles after the acute swimming challenge.

Serum Metabolites	LFD	HFD	LGG + HFD	^$^ OLP-01 + HFD	^$^ PL-02 + HFD	^$^ LY-66 + HFD
Lactate (nmol/L)_ before swimming (A)	3.81 ± 0.39	3.85 ± 0.59	3.85 ± 0.25	3.82 ± 0.32	3.86 ± 0.16	3.82 ± 0.20
Lactate (nmol/L)_10 min after swimming (B)	5.85 ± 0.44 ^###^	7.06 ± 0.65	6.31 ± 0.60 ^#^	5.53 ± 0.59 ^###^	5.20 ± 0.67 ^###^	5.11 ± 0.62 ^###^
Lactate (nmol/L)_rest after 20 min of swimming (C)	5.10 ± 0.23 ^###^	6.10 ± 0.34	5.34 ± 0.30 ^###^	4.81 ± 0.41 ^###^	4.44 ± 0.44 ^###^	4.42 ± 0.52 ^###^
Lactate production rate = (B/A)	1.54 ± 0.07 ^###^	1.85 ± 0.12	1.64 ± 07 ^###^	1.45 ± 0.04 ^###^	1.34 ± 0.13 ^###^	1.33 ± 0.10 ^###^
Lactate clearance rate = (B − C)/B	0.13 ± 0.04	0.13 ± 0.05	0.15 ± 0.04	0.13 ± 0.03	0.14 ± 0.05	0.13 ± 0.05
Glucose (mg/dL; 10 min after swimming)	130 ± 19	176 ± 11	175 ± 16	173 ± 14	178 ± 18	174 ± 12
Ammonia (NH_3_) (umol/L; 10 min after swimming) Decline rate of NH_3_ (%; compared to HFD)	137 ± 9 ^###^	165 ± 7	131 ± 10 ^###^	117 ± 9 ^###^	114 ± 7 ^###^	122 ± 8 ^###^
	16.84% ^###^	-	20.33% ^###^	29.14% ^###^	31.03% ^###^	20.33% ^###^
Creatine kinase (CK) (U/L; 10 min after swimming) Decline rate of CK (%; compared to HFD)	1345 ± 179 ^###^	1641 ± 163	1334 ± 172 ^###^	1140 ± 100 ^###^	921 ± 133 ^###^	902 ± 162 ^###^
	18.05% ^###^	-	18.71% ^###^	30.51% ^###^	43.88% ^###^	45.05% ^###^
BUN (mg/dL; 10 min after swimming) Decline rate of BUN (%; compared to HFD)	24.6 ± 1.2 ^###^	27.6 ± 1.2	24.3 ± 0.9 ^###^	22.2 ± 0.8 ^###^	22.0 ± 1.1 ^###^	21.7 ± 0.7 ^###^
	10.92% ^###^	-	11.92% ^###^	19.54% ^###^	20.35% ^###^	21.31% ^###^

^#^*p* < 0.05; ^###^
*p* < 0.001 compared with the high-fat diet control group. ^$^ OLP-01, PL-02, SA-03 and LY-66 were isolated from the same Olympic athlete.

## Data Availability

The datasets used and/or analyzed during the current study are available from the corresponding author on reasonable request.

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
