# Peer review of "Probiotic Strains Isolated from an Olympic Woman’s Weightlifting Gold Medalist Increase Weight Loss and Exercise Performance in a Mouse Model"

_nutrients, 2022, doi:10.3390/nu14061270_

Round 1
Reviewer 1 Report
Overall summary: This is an interesting paper that compares probiotic strains from healthy humans and an Olympic gold medal winner in weightlifting. The paper is generally well-written, flows well, and appears to be complete. I do have some concerns as outlined below, particularly with the importance attributed to extraction of probiotics from a single Olympic athlete in one event. Nonetheless, the manipulation of the human gut microbiome to improve health is an interesting and timely topic.
Major comments:
- Title: For more clarification, please add the sex and event of the athlete. Please clarify this in the paper as well.
- Why was this specific athlete chosen to examine probiotic bacteria?
- Are there any other indications that these species are common among elite weightlifters? Or is this really based on a case study? While the implication that there are probiotic strains that could help will weight loss and improve human health is intriguing, I question whether the source of the probiotic strains is the most important point of emphasis.
- Are there any indications that these probiotic strains can improve human health? In particular, is there any indication that the Olympic weightlifter who is the source of these strains exhibits a healthy gut?
- Why did the gut of the Olympic weightlifter manifest different strains than healthy individuals? Are there differences in diet that can account for this? Was the athlete taking any supplements that may have contributed to this?
- Methods: Why did the authors not include a group with a low-fat diet and probiotic supplementation? It seems unfair to compare a high-fat diet + probiotic supplementation to a low-fat diet group without supplementation. The paper would be far stronger with the addition of this group, and a better statistical approach could be applied.
- Figures 1: Please add the number of animals per group indicated by the data.
- For Figure 2, are these representative sections from each group? How many animals were there per group that were compared?
- Figure 3: How many experiments are represented by the data?
- Figure 4: How many animals per experimental group. Figure 4b: Muscle is misspelled.
- Discussion section: Is there any importance related to the fact that the probiotic strains were isolated from a weightlifting athlete? Would different results be expected with other athletes?
- Conclusion section: I feel that the paper needs a stronger conclusion. What specific future studies does the author suggest? Also, be more specific about how these findings may be applied for human health promotion. The current statement is rather vague.
Minor comments
- Line 42: BMI has a unit. Please add the unit kg/m-squared.
- The writing in the introduction is somewhat choppy and could be improved by better sentence-to-sentence transitions, including the use of appropriate transition/connector words/phrases.
- Line 54: “B. longum” should be italicized. Please make sure all bacterial genus/species combinations are italicized throughout the paper.
- Check the paper carefully for typos. There are quite a few (two spaces instead of one, etc.)
Author Response
Author's Reply to the Review Report (Reviewer 1)
Comments and Suggestions for Authors
Overall summary: This is an interesting paper that compares probiotic strains from healthy humans and an Olympic gold medal winner in weightlifting. The paper is generally well-written, flows well, and appears to be complete. I do have some concerns as outlined below, particularly with the importance attributed to extraction of probiotics from a single Olympic athlete in one event. Nonetheless, the manipulation of the human gut microbiome to improve health is an interesting and timely topic.
Major comments:
1.Title: For more clarification, please add the sex and event of the athlete. Please clarify this in the paper as well.
Response:
Thanks to your precious comment. I’ve edited the title to “Probiotic strains isolated from a Female Olympic Weightlifting gold medalist increase weight loss and exercise performance in a mouse model” according to your suggestion.
2. Why was this specific athlete chosen to examine probiotic bacteria?
Response:
Thanks to your precious comment. The female athletic Chen Wei-ling received weightlifting gold medal in 2008 Olympic game. She originally won the bronze medal in the 48 kg category, but later awarded the gold medal after the original gold and silver medalists were disqualified for drug use. During the time we collected and screened the various probiotic strain from various sources, the athlete Chen Wei-ling was also a PhD. student in the collaborative lab in Graduate Institute of Sports Science, National Taiwan Sport University, 33301 Taoyuan City, Taiwan (R.O.C.).
It’s a regular probiotic strains screening project and we didn’t purposely design the project to examine the probiotic function, which isolated from an Olympic gold medalist. But our research team coincidentally ran into the female athletic Chen Wei-ling while discussing collaborative program in Graduate Institute of Sports Science in National Taiwan Sport University and decided to isolate her with other health humans’ probiotic strains. Also, at first we didn’t precondition or expect that the probiotic strains isolated from the female gold medalist, Chen Wei-ling, would present better exercise performance to other healthy human donors’ strains in animal study in the beginning.
3. Are there any other indications that these species are common among elite weightlifters? Or is this really based on a case study? While the implication that there are probiotic strains that could help will weight loss and improve human health is intriguing, I question whether the source of the probiotic strains is the most important point of emphasis.
Response:
Thanks to your precious comment. Recently, studies began to investigate the probiotic supplementation influenced in athletes’ gut microbiome and athletic performance [1,2,3]. However, nearly non research carried on with and analyzed the different sources of isolated probiotic strains including elite athlete’s gut may bring in different impact on exercise performance. The present results are really based on a case study (probiotic strains isolated from the 2008 Olympic weightlifting gold medalist, Chen Wei-ling). The analysis data of other elite weightlifters or other sport-specific athletes is limited in this study and should be designed for further investigation in future.
The measurements of body fat, body weight and exercise performances belonged to our regular testing items and procedures of animal. Actually, the results of body fat and body weight could be counted as another individual article concerning weight loss. After through discussing with all authors, we decided to present and publish complete results including body fat, body weight and exercise performances to Nutrients.
We didn’t precondition or expect that the probiotic isolating sources would influence body lipid metabolism or exercise performances at first. By analyzing and ranking total animal data, it’s coincidental to discover that three probiotic strains (LY-66, PL-02, OLP-01) isolated from the 2008 Olympic weightlifting gold medalist- Chen Wei-ling exerted excellent exercise performance along with weight loss effect in animal model. The results of present study are interesting and surprising. But other factors including lifestyle, body composition, and dietary regime of the elite athlete’s may also regulate elite athletes’ gut microbiome as we discussed in the discussion part. However, the clinical trial of testing human exercise performance and lipid metabolism by taking different supplementations of probiotics isolated from various sport-specific elite athletes or strains from normal heathy human gut should be further investigated in future.
Ref.
1. Francavilla, V. C., Bongiovanni, T., Todaro, L., Di Pietro, V., & Francavilla, G. (2017). Probiotic supplements and athletic performance: a review of the literature. Med Sport, 70, 000-000.
2. Pyne, D. B., West, N. P., Cox, A. J., & Cripps, A. W. (2015). Probiotics supplementation for athletes–clinical and physiological effects. European journal of sport science, 15(1), 63-72.
3. Wosinska, L., Cotter, P. D., O’Sullivan, O., & Guinane, C. (2019). The potential impact of probiotics on the gut microbiome of athletes. Nutrients, 11(10), 2270.
4. Are there any indications that these probiotic strains can improve human health? In particular, is there any indication that the Olympic weightlifter who is the source of these strains exhibits a healthy gut?
Response:
Thanks to your precious comment. Previous study showed that
108 CFU of Lactobacillus. acidophilus TYCA-06 would significantly consume extra sugar (2% glucose or 6% monosaccharide) [1], which may benefit in glycemic homeostasis. The Lactobacillus plantarum PL-02 (isolated from the elite athlete) presented excellent anti-oxidative ability in DPPH assay [2]. Besides, the L. casei CS-773 and Streptococcus thermophilus SY-66 discovered their anti-pathogenic activity in fish [3]. Several studies had reported the Bifidobacterium longum subsp. Longum OLP-01 (isolated from the elite athlete) can improve exercise performance, regulate insulin resistance, assist obesity management and health promotion [4-8]. An animal study revealed that Lactobacillus salivarius subsp. salicinius SA-03 (isolated from the elite athlete) is capable of enhancing exercise performance and decreasing fatigue [9]. A animal study revealed the high dose treatment of Lactobacillus rhamnosus bv-77 can reduce body weight of HFD-Induced Obese Rats [10]. However, no report has demonstrated the human health promoting function of Lactococcus lactis subsp. lactis LY-66 (isolated from the elite athlete).
Four strains (OLP-01, PL-02, SA-03, and LY-66) isolated from Wei-Ling Chen, gold medal winner of the 2008 Summer Olympics women’s 48 kg weightlifting competition and the strain isolation process was performed under healthy condition of the donor (data not shown). But subjective questionnaire of lifestyle, body composition, and dietary regime of the elite athletes should be investigated in future.
Ref.
1. Hsieh, P. S., Ho, H. H., Hsieh, S. H., Kuo, Y. W., Tseng, H. Y., Kao, H. F., & Wang, J. Y. (2020). Lactobacillus salivarius AP-32 and Lactobacillus reuteri GL-104 decrease glycemic levels and attenuate diabetes-mediated liver and kidney injury in db/db mice. BMJ Open Diabetes Research and Care, 8(1), e001028.
- Lin, W. Y., Lin, J. H., Kuo, Y. W., Chiang, P. F. R., & Ho, H. H. (2022). Probiotics and their Metabolites Reduce Oxidative Stress in Middle-Aged Mice. Current Microbiology, 79(4), 1-12.
- Ho, H. H., Lu, C. L., Hsieh, P. S., Chen, C. W., Hsieh, S. H., Kuo, Y. W., ... & Lin, J. H. Y. (2022). Lactic acid bacteria metabolites in fish feed additives inhibit potential aquatic and food safety pathogens growth, and improve feed conversion. Journal of Applied Aquaculture, 1-21.
4. Lee, M. C., Hsu, Y. J., Chuang, H. L., Hsieh, P. S., Ho, H. H., Chen, W. L., ... & Huang, C. C. (2019). In vivo ergogenic properties of the bifidobacterium longum OLP-01 isolated from a weightlifting gold medalist. Nutrients, 11(9), 2003.
5. Lin, C. L., Hsu, Y. J., Ho, H. H., Chang, Y. C., Kuo, Y. W., Yeh, Y. T., ... & Lee, M. C. (2020). Bifidobacterium longum subsp. longum OLP-01 Supplementation during endurance running training improves exercise performance in middle-and long-distance runners: A double-blind controlled trial. Nutrients, 12(7), 1972.
6. Huang, W. C., Hsu, Y. J., Huang, C. C., Liu, H. C., & Lee, M. C. (2020). Exercise training combined with Bifidobacterium longum OLP-01 supplementation improves exercise physiological adaption and performance. Nutrients, 12(4), 1145.
7. Hsu, Y. J., Wu, M. F., Lee, M. C., & Huang, C. C. (2021). Exercise training combined with Bifidobacterium longum OLP-01 treatment regulates insulin resistance and physical performance in db/db mice. Food & Function, 12(17), 7728-7740.
8. Hsu, Y. J., Chiu, C. C., Lee, M. C., & Huang, W. C. (2021). Combination of Treadmill Aerobic Exercise with Bifidobacterium longum OLP-01 Supplementation for Treatment of High-Fat Diet-Induced Obese Murine Model. Obesity Facts, 14(3), 306-319.
9. Lee, M. C., Hsu, Y. J., Ho, H. H., Hsieh, S. H., Kuo, Y. W., Sung, H. C., & Huang, C. C. (2020). Lactobacillus salivarius subspecies salicinius SA-03 is a new probiotic capable of enhancing exercise performance and decreasing fatigue. Microorganisms, 8(4), 545.
10. Liao, C. A., Huang, C. H., Ho, H. H., Chen, J. F., Kuo, Y. W., Lin, J. H., ... & Yeh, Y. T. (2022). A Combined Supplement of Probiotic Strains AP-32, bv-77, and CP-9 Increased Akkermansia mucinphila and Reduced Non-Esterified Fatty Acids and Energy Metabolism in HFD-Induced Obese Rats. Nutrients, 14(3), 527.
5. Why did the gut of the Olympic weightlifter manifest different strains than healthy individuals? Are there differences in diet that can account for this? Was the athlete taking any supplements that may have contributed to this?
Response:
Thanks to your precious comment. Researchers has discovered that exercise would positively change the gut microbiota by [1]. Low intensity exercise altered the transient stool time and reduced contact time between the pathogens and the gastrointestinal mucus layer [2]. Besides, it’s proved that regular exercise may reduce inflammatory infiltrate and protect the morphology and the integrity of the intestine [3,4]
The long-term exercise regimen and strict diet of the Olympic weightlifting athlete may explain why their gut microbial species exhibited a more beneficial effect than those of nonprofessional athletes. However, the mechanism through which probiotic strains isolated from an Olympic medalist facilitate exercise performance and energy metabolism have yet to be fully elucidated
The elite athletes (in Taiwan) recieved qualify to participate Olympic games will be recruited and lived in the National Sports Training Center full-time (https://web.nstc.org.tw/English/). The National Sports Training Center will provide sticky exercise project and diet menu to every elite athlete (in Taiwan)
. The regular diet menu was listed as the follow link: https://www.nstc.org.tw/upload/FoodMenu/20220223211533628.pdf.
Ref:
1. Monda, V., Villano, I., Messina, A., Valenzano, A., Esposito, T., Moscatelli, F.,& Messina, G. Exercise modifies the gut micro-biota with positive health effects. Oxidative medicine and cellular longevity (2017), 2017:3831972.
- Bermon, S., Petriz, B., Kajeniene, A., Prestes, J., Castell, L., & Franco, O. L. (2015). The microbiota: an exercise immunology perspective. Exerc Immunol Rev, 21(21), 70-79.
3. Peters, H. P. F., De Vries, W. R., Vanberge-Henegouwen, G. P., & Akkermans, L. M. A. (2001). Potential benefits and hazards of physical activity and exercise on the gastrointestinal tract. Gut, 48(3), 435-439.
4. Campbell, S. C., Wisniewski, P. J., Noji, M., McGuinness, L. R., Häggblom, M. M., Lightfoot, S. A., ... & Kerkhof, L. J. (2016). The effect of diet and exercise on intestinal integrity and microbial diversity in mice. PloS one, 11(3), e0150502.
6. Methods: Why did the authors not include a group with a low-fat diet and probiotic supplementation? It seems unfair to compare a high-fat diet + probiotic supplementation to a low-fat diet group without supplementation. The paper would be far stronger with the addition of this group, and a better statistical approach could be applied.
Response:
Thanks to your precious comment. At present study, our major analysis strategy is to compare high-fat diet + probiotic supplementation to the high-fat diet group. We both compared high-fat diet + probiotic supplementation to the high-fat diet group and to the low-fat diet group in figure 1, Figure 4. We only compared high-fat diet + probiotic supplementation to the high-fat diet group in Table 2, Table 3. Previous study used exercise plus probiotic supplementation as a treatment to high-fat diet-induced obese murine model [1], in which author also compare high-fat diet + treatment group to the high-fat diet control group.
Ref:
Hsu, Y. J., Chiu, C. C., Lee, M. C., & Huang, W. C. (2021). Combination of Treadmill Aerobic Exercise with Bifidobacterium longum OLP-01 Supplementation for Treatment of High-Fat Diet-Induced Obese Murine Model. Obesity Facts, 14(3), 306-319.
7. Figures 1: Please add the number of animals per group indicated by the data.
Response:
Thanks to your precious comment. We will add the number of experimental animals per group to caption of Figure 1 (n=8 mice per group).
8. For Figure 2, are these representative sections from each group? How many animals were there per group that were compared?
Response:
Thanks to your precious comment. We dissected and stain tissue samples of each animal of the group (8 mice per group), then selecting representative sections from each group in Figure 2, as previously described [1,2].
Ref:
1. Kan, N. W., Lee, M. C., Tung, Y. T., Chiu, C. C., Huang, C. C., & Huang, W. C. (2018). The synergistic effects of resveratrol combined with resistant training on exercise performance and physiological adaption. Nutrients, 10(10), 1360.
2. Nichols, A. W. (2007). Probiotics and athletic performance: a systematic review. Current sports medicine reports, 6(4), 269-273.
9. Figure 3: How many experiments are represented by the data?
Response:
Thanks to your precious comment.
In Figure 3, all experiments were done in triplicate.
10. Figure 4: How many animals per experimental group. Figure 4b: Muscle is misspelled.
Response:
Thanks to your precious comment. In Figure 4, all experiments were done in triplicate. I’ve corrected the misspelled word “Muscle” here.
11. Discussion section: Is there any importance related to the fact that the probiotic strains were isolated from a weightlifting athlete? Would different results be expected with other athletes?
Response:
Thanks to your precious comment. I will add the relation of the fact that the probiotic strains were isolated from a weightlifting athlete (Wei-Ling Chen, the 2008 Summer Olympics women’s 48 kg weightlifting gold medalist) to discussion section. And add the limitation paragraph to discussion part:
“However, whether the probiotic strains isolated from other sport-specific elite athletes could also improve exercise performance should be tested and analyzed in the future.”
12. Conclusion section: I feel that the paper needs a stronger conclusion. What specific future studies does the author suggest? Also, be more specific about how these findings may be applied for human health promotion. The current statement is rather vague.
Response:
Thanks to your precious comment.
I will strengthen the structure of the conclusion section and relate some more specific statements including how these findings may be applied for human health promotion according to your suggestion. The edited conclusion section is as follows:
“The findings of this study could be applied in weight management, lipid metabolism and exercise for the promotion of human health. Probiotic strains may involve lipid metabolism via absorbing excess fatty acids in the gastrointestinal tract. Clinical experiments including life-style questionnaire, diet record, body weight, BMI, health condition, aerobic exercise, anaerobic exercise, serum biochemistry profile, gut microbiota analysis and metabolomics analysis by taking probiotic strains of OLP-01, PL-02, and LY-66 should be tested in the future.”
Minor comments
1. Line 42: BMI has a unit. Please add the unit kg/m-squared.
Response:
Thanks to your precious comment. I will edit the unit of BMI (kg/m2) at line 42.
2. The writing in the introduction is somewhat choppy and could be improved by better sentence-to-sentence transitions, including the use of appropriate transition/connector words/phrases.
Response:
Thanks to your precious comment. I will edit and modify the introduction section with appropriate transition/connector words/phrases.
3. Line 54: “B. longum” should be italicized. Please make sure all bacterial genus/species combinations are italicized throughout the paper.
Response:
Thanks to your precious comment. I will thoroughly check the all bacterial genus/species mentioned in this article to be italicized form.
4. Check the paper carefully for typos. There are quite a few (two spaces instead of one, etc.)
Response:
Thanks to your precious comment. I will thoroughly check and correct typos in this article. Besides, the English grammar of this article has been edited and enhanced by Wallace Academic Editing.
Reviewer 2 Report
Comments:
- Could the authors please clarify how the numbers of animals for their experiments were determined i.e. how was their study powered?
- Could the authors please justify the use of parametric statistical analyses?
Author Response
Author's Reply to the Review Report (Reviewer 2)
Comments and Suggestions for Authors:
1.Could the authors please clarify how the numbers of animals for their experiments were determined i.e. how was their study powered?
Response:
Thanks to your precious comment. Wan Nor Arifin et al brought a simplified equation to calculate sample size in animal studies [1]. According to Arifin’s equation, the estimated animal size per group would be:
10/ kr+1 to 20/ kr + 1 (K= numbers of groups, r= repeat measure times)
(10/ 18*2+ 1) to (20/18*2 +1)
It’s around 1 to 2 mice /per group. However, if we need to sacrifice animals at end of the study, animal size need to times 3, which become 3 to 6 mice/ per group. Besides, we estimated 2 outliers in each group, therefore it will need 3+2 to 6+2 mice per / group.
Ref.
1. Arifin, 2017. Sample Size Calculation in Animal Studies Using Resource Equation
Approach. Malays J Med Sci 24:101.
2.Could the authors please justify the use of parametric statistical analyses ?
Response:
Thanks to your precious comment. The D'Agostino-Pearson normality test was used to justify the normality of the data (Graphpad Prism 8, Graphpad Software, San Diego, CA, USA). Normality testing results showed that the data in each group are normal (probabilities > 0.05).

Round 2
Reviewer 1 Report
The authors have appropriately addressed my comments.
Author Response
Response:
Thanks to your kindly and precious reply. I will check English spell in this article again.
Reviewer 2 Report
The authors have satisfactorily responded to my comments.
Author Response
Comments and Suggestions for Authors
The authors have satisfactorily responded to my comments.
Response:
Thanks to your kindly and precious reply.
